# Targeting Tumor Microenvironment Interactions in Chronic Lymphocytic Leukemia Using Leukotriene Inhibitors

**DOI:** 10.3390/ijms26052209

**Published:** 2025-02-28

**Authors:** Laia Sadeghi, Magali Merrien, Magnus Björkholm, Anders Österborg, Birgitta Sander, Hans-Erik Claesson, Anthony P. H. Wright

**Affiliations:** 1Department of Laboratory Medicine, Division of Biomolecular and Cellular Medicine, Karolinska Institutet, 17177 Stockholm, Sweden; anthony.wright@ki.se; 2Department of Laboratory Medicine, Division of Pathology, Karolinska Institutet, 17177 Stockholm, Sweden; magali.merrien@ki.se (M.M.); birgitta.sander@ki.se (B.S.); 3Department of Medicine Solna, Karolinska Institutet, 17177 Stockholm, Sweden; magnus.bjorkholm@ki.se (M.B.); hans-erik.claesson@ki.se (H.-E.C.); 4Department of Oncology-Pathology, Karolinska Institutet, 17176 Stockholm, Sweden; anders.osterborg@ki.se; 5Department of Hematology, Karolinska University Hospital, 17176 Stockholm, Sweden

**Keywords:** chronic lymphocytic leukemia, 5-LOX pathway, BTKi, MK886, zileuton, tumor microenvironment, co-culture

## Abstract

Chronic lymphocytic leukemia (CLL) cells depend on microenvironment niches for proliferation and survival. The adhesion of tumor cells to stromal cells in such niches triggers the activation of signaling pathways crucial for their survival, including B-cell receptor (BCR) signaling. While inhibitors of Bruton’s tyrosine kinase (BTKi) have shown efficacy in patients with CLL by disrupting these interactions, acquired resistance and toxicity remain a challenge during long-term therapy. Thus, identifying additional therapeutic modalities is important. Previously, we demonstrated that 5-lipoxygenase (5-LOX) pathway inhibitors reduced mantle cell lymphoma (MCL) cell adhesion to stromal cells, motivating us to investigate their potential in the context of CLL. We employed an ex vivo co-culture model to study CLL cell adhesion to stromal cells in the absence and presence of 5-LOX pathway inhibitors (zileuton and MK886) as well as the BTKi ibrutinib that was included for comparative purposes. Our findings demonstrated that different CLL samples adhere to stromal cells differentially. We observed a variable decrease in CLL cell adhesion to stromal cells following the inhibition of the 5-LOX pathway across a spectrum of patient samples that was distinct to the spectrum for ibrutinib. Positive and negative correlations were shown between the clinical and genetic features of the CLL samples and their level of adherence to stromal cells in both the absence and presence of the tested inhibitors. These results suggest the 5-LOX pathway as a candidate for assessment as a new therapeutic target in CLL.

## 1. Introduction

Chronic lymphocytic leukemia (CLL) is the most common leukemia in adults in the USA and Europe including the Nordic countries [1]. Each year, approximately 20,700 new cases are diagnosed, with 4440 deaths reported across both genders [2]. It typically occurs in elderly patients and is characterized by the clonal proliferation and accumulation of CD5 positive B cells within the blood, bone marrow, lymph nodes, and spleen [3,4]. CLL displays a heterogeneous clinical progression among patients, ranging from early-stage disease that does not require therapy to a more aggressive disease that requires immediate treatment [5,6,7]. CLL treatment has evolved from chemoimmunotherapy to small-molecule-based precision therapies like BTK inhibitors (ibrutinib and others) and BCL-2 inhibitors (venetoclax) that significantly improve patient outcomes, with CAR-T cells and other new therapeutics as emerging options [4,7,8]. Multiple factors including age, cytogenetic aberrations, and the mutational status of the immunoglobulin heavy chain variable region (*IGHV*) as well as other genomic/chromosomal abnormalities, del(17p)/*TP53* mutation in particular, are used to define optimal therapeutic strategies [5,9,10,11,12,13,14].

The tumor microenvironment plays an essential role in the development, survival, and drug resistance of CLL cells [15,16]. CLL cells proliferate and survive in the bone marrow and lymph node microenvironments, where they receive growth and survival signals from various stromal cells that provide survival signals [17,18,19]. The adhesion of CLL cells to stromal cells is mediated by multiple adhesion molecules including β1 and β2 (beta1 and beta2) integrins, BCR signaling, and the cell surface receptor CD44, where the expression levels of these molecules affect CLL cell migration and adhesion [20,21]. Despite the development of modern targeted therapies, CLL is still regarded as an incurable disease with a yet unmet clinical need in patients with double-refractory (BTK and BCL-2) disease [22,23].

The molecule 5-lipoxygenase (5-LOX) catalyzes the conversion of arachidonic acid to leukotrienes (LTs), which are proinflammatory mediators initiating allergy responses [24,25]. Upon cell activation, 5-LOX translocates to the nuclear membrane where it interacts with the 5-LOX activation protein (FLAP), mediating the generation of LTs [26]. A FLAP inhibitor (MK886) and a 5-LOX inhibitor (zileuton) both reduce leukotriene synthesis, airway inflammation, and allergic asthmatic responses [27]. Several studies have shown that certain B-cell lymphomas and CLL cells have abnormally high levels of 5-LOX expression, which is associated with advanced forms of cancer [28,29]. The biological function of the 5-LOX pathway in B cells is unclear, but recent studies indicate that it may have a role in the migration and adherence of malignant B cells [30]. The inhibition of the 5-LOX pathway in the mantle cell lymphoma (MCL) cell line using zileuton and GSKi (FLAP inhibitor) decreased MCL cell migration towards CXCL12 (a ligand for the CXCR4 receptor) and reduced their stromal cell adhesion [31]. Moreover, leukotriene B4 with CXCL12 synergistically induces the migration of lymphoma cells to the lymph node in DLBCL (diffused large B-cell lymphoma) [32]. Since CLL shares many characteristics with MCL, it is of interest to determine whether the inhibition of the 5-LOX pathway reduces the stromal cell adhesion of CLL cells [33].

Here, we investigated the effects of inhibiting the 5-LOX pathway using zileuton and MK886 on the stromal cell adhesion of CLL cells in an ex vivo co-culture system. Additionally, we explored whether patient mutation status could predict the efficacy of these inhibitors in modulating CLL cell adhesion to stromal cells.

## 2. Results

### 2.1. Characterization of CLL Cells Isolated from Patients

Patients under study were seen at the Hematology Outpatient Clinic of Karolinska University Hospital. CLL cells from blood were obtained from ten patients (five males and five females), with a median age of 56.5 years (range 34–79 years) (Table 1). No patient was under active anticancer therapy at sampling; seven patients were previously untreated at sampling, while three had a treatment-free interval of at least one year preceding sampling. All patients had a blood lymphocyte count > 30 × 10^9^/L, with a median of 107 × 10^9^/L (range 35–500 × 10^9^/L). This enabled us to have enough cells for the experiments.

We performed Principal Component Analysis (PCA) to reduce data complexity and explore the relationships between the CLL patient samples used in this study based on their mutation status (Figure 1a) [34]. The PCA demonstrated significant heterogeneity among the patient samples, with some samples showing similarity to each other. Specifically, patients #182, #183, and #093 exhibited clustering tendencies, perhaps correlating with the deletion of the 13q;14 chromosomal region and the presence of *IGHV* mutations in all three samples. Additionally, the PCA plot shows that patients #152 and #053 form a distinct cluster, which is associated with their lack of *IGHV* mutations. In conclusion, the selected samples reflect, at least in part, the clinical heterogeneity of CLL, but combinations of particular mutations appear to indicate subgroups in which samples are more similar to each other.

### 2.2. Heterogeneous Stromal Cell Adhesion Activity of Patient-Derived CLL Cells Ex Vivo

To assess the adhesion capacity of the CLL samples, fluorescently labeled CLL cells were co-cultured with a mono-layer of fluorescently labeled HS-5 stromal cells for 4 h, and the ratio of bound CLL cells to stromal cells was then measured using flow cytometry, as exemplified in Appendix A; other studies have also used flow cytometry for similar assessments [35]. The JeKo-1 MCL cell line, known to adhere to HS-5 stromal cells, was used as a control, as demonstrated in our previous studies [36,37]. The data showed that while CLL cells from all ten patients adhered to stromal cells, there were noticeable differences in the level of adherence between samples (Figure 1b). Specifically, leukemic cells from two CLL patients (#54 and #93) exhibited consistently higher adherence compared to other samples and the JeKo-1 cells. In contrast, five samples (#53, #88, #172, #182, and #183) exhibited consistently lower adhesion levels. The remaining three samples showed inconsistent adherence levels between replicates and were thus more difficult to interpret. Differences in cell viability between replicates (Appendix A) could not account for differences in adhesion levels. Interestingly, the separation of the samples in the PCA1 dimension in Figure 1a showed a degree of association with adhesion levels, and samples with lower adhesion tended to have higher PCA1 values. This finding suggests a potential link between the ex vivo adhesion level of CLL cells and the mutation status of the patients.

### 2.3. Heterogeneous Effect of 5-LOX Pathway Inhibitors on Stromal Cell Adhesion Activity of Patient-Derived CLL Cells Ex Vivo

Next, we evaluated the effects of 5-LOX pathway inhibitors on the adhesion levels of the patient-derived CLL cells to stromal cells. The ex vivo adhesion assay was performed in the presence and absence of the 5-LOX inhibitor zileuton and the FLAP inhibitor MK886 (Figure 2 and Appendix A). Ibrutinib was used as a control as BTKis are known to affect CLL adhesion to stromal cells by promoting the detachment of CLL from stromal cells in tumor microenvironments [38,39,40].

The impact of ibrutinib on leukemia cell adhesion varied, with a significant reduction observed in the adhesion level in four out of ten samples (Figure 2a). In contrast, MK886 and zileuton caused significant reductions in adhesion in most of the samples (Figure 2b,c). For MK886, the only sample that did not reach statistical significance was #166 (*p*-value = 0.074), likely due to a high variance in the measured data rather than a lack of an effect on adherence. The zileuton treatment significantly reduced adhesion levels in six out of ten samples, all of which were significantly affected by MK886 (Figure 2b,c).

To further assess the extent of similarity of the drug–effect patterns for the different inhibitors, we made pairwise comparisons and calculated Pearson’s correlation coefficients for each pair of values, using the ratio of treated to untreated values as explained in [41] (Figure 2). The results show that, while there is a poor or no correlation between the effects of ibrutinib and either of the 5-LOX pathway inhibitors, there is a moderately strong correlation (r = 0.75) between the effects of MK886 and zileuton (Figure 2d–f). We conclude that there is heterogeneity in the sensitivity of the different CLL samples to the drugs tested, and the effects of the 5-LOX pathway inhibitors resemble each other to a greater extent than to ibrutinib. Furthermore, the 5-LOX pathway inhibitors appear to affect the adhesion of CLL cells to stromal cells in a higher number of samples compared to ibrutinib (Figure 2b,c).

### 2.4. Mutation Status and Lymphocyte Count for CLL Samples Are Associated with the Level of Their Adhesion to Stromal Cells in the Absence and Presence of Small-Molecule Inhibitors

PCA (Figure 3a), evaluating mutation variables together with adhesion levels in the absence and presence of the 5-LOX pathway inhibitors (MK886 and zileuton) or ibrutinib, shows differences in observed clustering patterns compared to those seen for mutation variables alone (Figure 1a). A notable cluster (#093, #182, and #183) based on only mutation variables is dispersed when cell adhesion data are included in the PCA, indicating that while these patients share some common mutation status characteristics (e.g., the presence of mutations in *IGHV* and the 13q;14 deletion), their cell adhesion levels in the absence and presence of inhibitors are not sufficiently similar for the cluster to be seen when adhesion data were added to the analysis. Consistently, cells from sample #093 exhibit much higher adhesion activity but lower ibrutinib sensitivity than those from samples #182 and #183. Of these three samples, #183 clusters closely to samples #053 and #152, as shown in Figure 3a. Samples #088 and #172 cluster quite closely based on mutation variables alone, perhaps because they are from patients that are both relatively low in age and are the only patients exhibiting trisomy 12, but they appear to be more closely clustered when adhesion data were added. This could be because they both exhibit low cell adhesion levels that are not significantly reduced by ibrutinib but are sensitive to MK886.

To better understand the relationship between mutation status and other clinical features of CLL samples with the effects of inhibitors on adhesion, we performed a cross-correlation analysis (Figure 3b). Interestingly, the level of adhesion of CLL to stromal cells shows a fair, moderately strong positive, or negative correlation to all but one of the variables tested. Correlation *p*-values are shown in Appendix A. The adhesion level shows moderately strong positive correlations (r = 0.65) to lymphocyte count, 17p deletion (r = 0.65, *p*-value = 0.03), and the presence of mutations in *IGHV* (r = 0.64, *p*-value = 0.04). Fair correlations with the adhesion level are seen for the deletion of 13q;14 (positive) (r = 0.46, *p*-value = 0.04), trisomy 12 (negative) (r = −0.37, *p*-value = 0.08), and the deletion of 11q (negative) (r = −0.29, *p*-value = 0.08). For cells treated with zileuton, there is a very strong negative correlation between their adhesion level and the presence of the 11q deletion. This negative correlation is shown more weakly for the other inhibitor of the 5-LOX pathway, MK886, but not for ibrutinib. This could indicate a specific sensitivity of CLL cells deleted for 11q to the 5-LOX pathway inhibitors, but it should be noted that only one of the samples bears the 11q deletion. Contrastingly, the adhesion level of cells in the presence of zileuton shows a moderately strong positive correlation (r = 0.65, *p*-value = 0.02) with the presence of the 13q;14 deletion, indicating that the presence of this deletion might be associated with a reduced efficacy of zileuton. No similar correlation is seen for cells treated with ibrutinib, and thus, this association may be specific for the inhibitors of the 5-LOX pathway. A similar association may also apply for CLL cells mutated in *IGHV,* for which a fair correlation was observed, but in this case, a similar positive correlation was also seen for cells treated with ibrutinib. Only weak correlations were found between the level of adhesion in the presence of MK886 and the mutation variables studied. This may be due in part to the broader spectrum of efficacy observed for MK886, compared to zileuton or ibrutinib.

## 3. Discussion

In this pilot study, we examined leukemic cells from ten patients with CLL in relation to their capacity for adhesion to stromal cells and the sensitivity of this adhesion to inhibitors of the 5-LOX pathway (zileuton and MK886). Ibrutinib, a known inhibitor of lymphoma cell adhesion to stromal cells, was included for comparative purposes [36,40,42]. Although a relatively small number of patient samples were studied, the samples showed heterogeneity of mutation status, as expected for CLL, which is known to be a highly heterogeneous malignancy [43]. Interestingly, the adhesion capacity measured for the samples was also highly heterogeneous and showed correlations with molecular and clinical features. There was also heterogeneity in the sensitivity of adhesion to 5-LOX inhibitors, with sensitivity patterns that differed to the sensitivity pattern for ibrutinib.

The positive correlations observed between molecular and clinical features and the level of adhesion to stromal cells ex vivo are not sufficient to support conclusions about causality, but it is worthwhile to speculate about how adhesion level might be functionally related to clinical biological features. If supplanted to an in vivo context, high adhesion to stromal cells might indicate a higher capacity of such CLL cells to occupy microenvironment niches such as the bone marrow or lymph node, where leukemic cells are known to receive signals that foster their proliferation and survival [44]. This might lead to an increased accumulation of cancer cells and consequently higher lymphocyte counts in the bloodstream. One result of the p17 deletion is the deletion of the *TP53* gene encoding the p53 tumor suppressor protein [45,46]. *TP53* plays an important role in maintaining the genomic integrity of cells as well as in the apoptotic disposal of damaged cells; therefore, it is easy to see how the deletion of *TP53* could facilitate the evolution of CLL cells, including the acquisition of adaptations that enhance their capacity for adhesion to stromal cells [47]. Finally, the association of adhesion capacity with *IGHV* mutations suggests that CLL cells exhibiting high adhesion capacity may originate from B cells later in B-cell development that have undergone a hypermutation of the immunoglobulin heavy chain variable regions, a characteristic of the milder biological behavior associated with mutated *IGHV* [48].

It appears that 5-LOX pathway inhibitors have an inhibitory effect on the adhesion of CLL cells to stromal cells, albeit to varying degrees for different samples, and the pattern of efficacy for the different samples differs to that of ibrutinib. The varying efficacy of the inhibitors correlates to some of the molecular and clinical features. For example, the presence of mutations in *IGHV* shows a fair level of correlation with the levels of adhesion in the presence of both zileuton and ibrutinib, suggesting that CLL cells derived from more highly developed B cells tend to be less sensitive to these drugs. However, other mutations or factors, which were not addressed in this study, may also influence drug sensitivity and adhesion levels, potentially in combination with *IGVH* mutations. More specifically, for 5-LOX pathway inhibitors, the results suggest that the 13q;14 deletion is associated with a lower efficacy of these drugs (particularly zileuton), while the 11q deletion appears to be associated with higher efficacy, though this is accompanied by high variability in zileuton responses. The latter, though, underlies the need for larger studies also since the very strong negative correlation of the presence of the 11q deletion with reduced adherence in the presence of zileuton is based on only one of the ten samples bearing this deletion.

While the specifics of the correlations seen may be uncertain in some samples, the results do clearly show that 5-LOX pathway inhibitors do show a different spectrum of efficacy compared to ibrutinib. As such 5-LOX inhibitors do warrant further study to clarify the molecular mechanisms of action that distinguish them from BTKi and to assess their suitability as a complementary alternative to ibrutinib family drugs. If they are shown to be appropriate, they could be used to treat patients that respond poorly to BTKi or that become resistant to them during therapy. Under such circumstances, an advantage would be that some 5-LOX pathway inhibitors, such as zileuton, are already clinically approved for the treatment of other indications [49,50]. However, the main limitation of this study was the cohort size, as we analyzed only ten patients. Nevertheless, the diverse mutation profile of these patients adequately captured the heterogeneity of CLL, supporting the relevance of our findings for larger cohort studies.

## 4. Materials and Methods

### 4.1. Cell Lines and Reagents

The JeKo-1 and HS-5 stromal cells were cultivated at 37 °C and 5% CO_2_ in media supplemented with 100 U/mL penicillin and 100 μg/mL streptomycin. The HS-5 human stromal cell line was obtained from ATCC (American Type Culture Collection, Manassas, VA, USA). The MCL cell line JeKo-1 was purchased from DSMZ (German Collection of Microorganisms and Cell Culture GmbH, Braunschweig, Germany). The HS-5 and JeKo-1 cell lines were maintained in DMEM (Gibco, NY, USA) or RPMI–glutamax (Gibco), respectively, supplemented with 10% heat-inactivated fetal bovine serum (HI FBS, Gibco). The co-cultures of JeKo-1 and HS-5 or CLL cells and HS-5 at a 10:1 ratio were maintained under the same conditions as for HS-5 cells alone.

### 4.2. Patients Blood Samples and CLL Cell Enrichment

CLL cells from blood were obtained after signed informed consent according to the Declaration of Helsinki and approved by National Ethics Authority (www.etikprovningsmyndigheten.se, accessed on 10 November 2021). All patients included in this study were diagnosed with CLL in the region of Stockholm (Karolinska University Hospital). Blood samples were collected in EDTA (ethylenediaminetetraacetic acid), and CLL cells were enriched by negative selection using RosetteSep™ according to the manufacturer’s protocol (StemCell Technologies, Vancouver, BC, Canada). The cells were then collected using Ficoll-Paque PLUS (GE Healthcare Life Science, Chicago, IL, USA), washed with PBS two times, and directly frozen in 50% RPMI-GM, 40% FBS, and 10% DMSO (preserved at −150 °C) [51]. Frozen isolated CLL cells were thawed and kept in 20% FBS in RPMI for 1h and then used for functional assays.

### 4.3. Cell–Cell Binding Assay and Flow Cytometry

CellTracer Carboxyfluorescein succinimidyl ester (CFSE, C34554 ThermoFisher, Waltham, MA, USA) and Far-red (C34564 ThermoFisher, Waltham, MA, USA) were used to label lymphoma cells (JeKo-1 or primary CLL cells) and HS-5 stromal cells, as described fully in [36]. Both CFSE and Far-red titrations were performed to ensure optimal concentration. HS-5 stromal cells were stained with Far-red and conditioned for 24 h before co-culturing with leukemia cells. Briefly, after staining, CLL cells were added to monolayers of HS-5 cells and co-cultured for 4 h. Next, unbound or loosely bound CLL cells were removed by washing twice with room temperature PBS. The bound cells, together with stromal cells, were harvested using trypsin and then subjected to flow cytometry. After data acquisition, gating was applied to exclude cell debris and dead cells using DAPI (F6057, Sigma-Aldrich, St. Louis, MO, USA) staining. Large cell aggregates were also excluded during the gating process. Subsequently, gating based on CFSE and Far-red staining was performed to separately identify CLL cells from stromal cells. The number of events within each gate was determined, and the normalized level of adherent cells was calculated as events in the CFSE gate divided by events in the Far-red gate. Flow cytometry data were obtained using a MACSQuant Analyzer 10 (Miltenyi Biotec, Bergisch Gladbach, Germany) and analyzed using the FlowJo software (FlowJo version 10).

### 4.4. Statistical Analysis

All co-culture assays were conducted with at least four technical replicates. Results were presented as box plots displaying the first quartile, third quartile, and median values. Data normality was assessed using the Shapiro–Wilk test, which is a statistical test that evaluates whether a dataset follows a normal distribution using *p*-values. The *p*-value from the test was *p* < 0.05, suggesting that the data does not follow a normal distribution. For paired comparisons between the treated and untreated (Ctrl) samples, the Wilcoxon signed-rank test was employed. Principal Component Analysis (PCA) was performed on the mutation statuses of included patient samples, alone or together with collected adhesion data, after scaling. All mutation statues were included in the analysis (*IGHV* mutation status, del (17p), del (13q;14), del (11q), and Trisomy 12). Cross correlation analysis (Pearson) was used to compare adhesion data with the mutation statuses of included patient samples and presented using ggplot2 (version 3.5.1) for visualization as a heatmap. Mutation statuses were categorized, with “0” indicating the absence of a genetic aberration and “1” indicating its presence. Lymphocyte count was categorized as “low” (<89 × 10^9^/L), “medium” (89 to 166 × 10^9^/L), or “high” (>166 × 10^9^/L) in order to create similarly sized categories. As suggested elsewhere, the strength of correlations was described as “very strong” (r ≥ 0.8), “moderately strong” (r = 0.6 to 0.79), “fair” r = 0.3 to 0.59), and poor (r < 0.3) [12].

## 5. Conclusions

CLL cells from patients exhibited heterogeneous ex vivo adhesion patterns to stromal cells and demonstrated distinct sensitivity profiles to 5-LOX inhibitors compared to the BTKi ibrutinib. Notably, the variability in sensitivity to the 5-LOX pathway inhibitors appeared to correlate to the mutation status of the patient samples. This study provides a starting point for further studies aimed at understanding the molecular mechanisms that distinguish the 5-LOX pathway inhibitors from BTKi as well as assessing whether 5-LOX inhibitors might be a potential treatment of CLL.

## Figures and Tables

**Figure 1 ijms-26-02209-f001:**
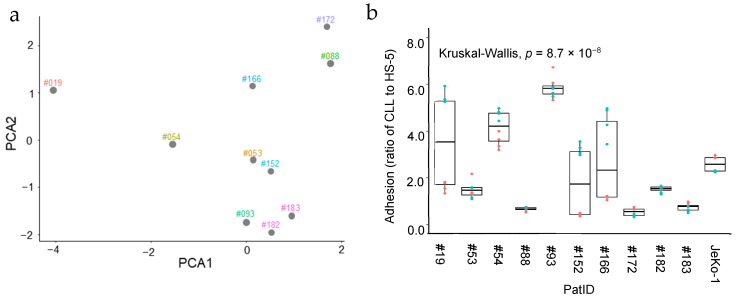
Heterogeneous mutation status and stromal cell adhesion capacity of patient-derived CLL cells ex vivo. (**a**) Principal Component Analysis (PCA) illustrating the distribution of CLL patient samples based on mutation status (*IGHV* mutation status, del (17p), del (13q;14), del (11q), and Trisomy 12), shown in Table 1. (**b**) Box plot showing the level of adhesion to stromal cells for CLL cells from ten patient samples using a cell–cell binding assay. The patient-derived CLL cells were cultured with a mono-layer of HS-5 stromal cells for 4 h prior to the assay. The JeKo-1 MCL cell line was included as a positive control. Experiments were performed in triplicate for two biological replicates, with point colors representing the different replicates. The Kruskal–Wallis test was used to assess whether there was a significant difference in the adhesion levels between the samples.

**Figure 2 ijms-26-02209-f002:**
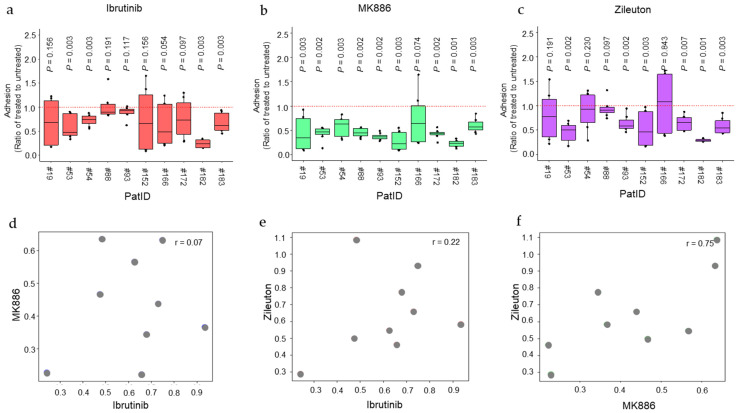
Primary CLL cells from different patients respond differently to small-molecule inhibitors. (**a**–**c**) Adhesion levels of CLL cells from different samples to stromal cells were assessed using a cell–cell binding assay in the absence and presence of 0.5 μM ibrutinib and 1 μM MK886 or 1 μM zileuton. Box plots illustrate the median value of data normalized to values for untreated cells, and individual values outside the central quartiles are shown. The hashed red line represents the normalized value for untreated cells. Statistical significance between the treated and untreated conditions was evaluated using the Wilcoxon test, with *p*-values indicating the levels of significance for differences. Experiments were performed in four technical replicates for each of the two independently conducted experiments. (**d**–**f**) Pearson’s correlation analysis was performed to compare the median effect of ibrutinib with MK886 (**d**), ibrutinib with zileuton (**e**), and MK886 with zileuton (**f**) on the adhesion levels. The Pearson correlation coefficient (r) represents the strength of the linear relationship between the effects of the treatments.

**Figure 3 ijms-26-02209-f003:**
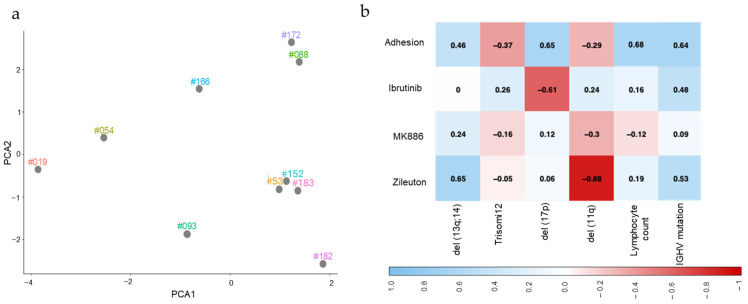
The mutation status of CLL samples is associated with the level of their adhesion to stromal cells in the absence and presence of small-molecule inhibitors (**a**) PCA clustering was performed using both experimental data (adhesion values calculated as the ratio of CLL cells to stromal cells in the treated and untreated samples) and mutation status (*IGHV* mutation status, del (17p), del (13q;14), del (11q), and Trisomy 12), as shown in Table 1. (**b**) A cross-correlation analysis was performed to evaluate the level of association between the studied variables and the capacity of lymphoma cells to adhere to stromal cells without added drugs and in the presence of 0.5 µM ibrutinib, 1 µM MK886, and 1 µM zileuton. In the heatmap, blue indicates a positive correlation, while red represents a negative correlation.

**Table 1 ijms-26-02209-t001:** Molecular and clinical features of CLL samples. This table summarizes the molecular and clinical features of the CLL patients included in this study at the time of diagnosis. Features include age, sex, lymphocyte count, and the presence of specific genetic aberrations. “+” represents the presence of a mutation/deletion, and “−” indicates its absence.

Patient ID	Sex	Age	Lymphocyte Count (10^9^ /L)	*IGHV* Mutation Status	del (17p)	del (13q;14)	del (11q)	Trisomy 12
#182	F	73	87.5	+	−	+	−	−
#183	F	59	66	+	−	+	−	−
#019	F	79	500	−	+	−	−	−
#053	F	58	100	−	−	−	+	−
#054	M	75	286	−	−	+	−	−
#088	M	44	35	−	−	−	−	+
#093	F	78	114	+	−	+	−	−
#152	M	72	70	−	−	+	−	−
#166	M	47	260	−	+	+	−	−
#172	M	34	166	+	−	−	−	+

## Data Availability

The patient information and mutation status that support the finding of this study is available in Table 1. The authors confirm that the data supporting the findings of this study are available within the article and its Appendix A.

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
