# Peer review of "Targeting Tumor Microenvironment Interactions in Chronic Lymphocytic Leukemia Using Leukotriene Inhibitors"

_ijms, 2025, doi:10.3390/ijms26052209_

Round 1
Reviewer 1 Report
Comments and Suggestions for Authors
Targeting tumor microenvironment interactions in chronic lymphocytic leukemia using leukotriene inhibitors. Here is some minor queries
Comments:
1. In the second paragraph, the authors mentioned the role of tumor microenvironments in cancer progression in a limited way. My suggestion to the authors is to elaborate on this section, as it must focus on the role of TME in adhesion and other extracellular matrices in the disease progression.
2. The patient sample size is very small, with only 10 patients with heterogeneity. How do we reach any conclusions? I am concerned about this.
3. Is there age and other comorbidity correlated with the adhesion capacity of CLL patient samples?
4. Is there any relationship between mutation in CLL patients and adhesion phenotypes? This means certain mutations make more adhesive cells, as we saw in patients 19 and 166. If it does show, please elaborate in the discussion section.
5. Please add the limitations of this study in the last paragraph of the discussion.
6. Are there any co-culture fluorescent image authors taken for both normal and disease conditions, just like Figure S1b?
Author Response
The authors thank the reviewer for their time and valuable comments, which will help improve our manuscript. We have addressed the reviewer’s questions and comments as outlined below
- In the second paragraph, the authors mentioned the role of tumor microenvironments in cancer progression in a limited way. My suggestion to the authors is to elaborate on this section, as it must focus on the role of TME in adhesion and other extracellular matrices in the disease progression.
The text was added from lines 51 to 54, including references 20 and 21
- The patient sample size is very small, with only 10 patients with heterogeneity. How do we reach any conclusions? I am concerned about this.
Thank you for the comment. The main overall conclusions that are well supported by the data are that CLL cells from different patients are heterogeneous in the stromal cell adhesion activity and that adhesion levels are differentially affected by 5-LOX pathway inhibitors in a pattern that differs from the pattern for ibrutinib. There are also significant associations between these characteristics and genetic markers but here the extra stratification needed reveals the limitations of this small patient group. We see this is an exploratory study that is sufficient to motivate further ex vivo studies using patient samples. However, we fully acknowledge that further comprehensive investigations are necessary to gain a deeper understanding of the underlying mechanisms underlying our main conclusions.
- Is there age and other comorbidity correlated with the adhesion capacity of CLL patient samples?
There is no clear association of age with adhesion activity but as the reviewer has pointed out, the data set is very small and we see this as a question to be addressed by larger studies. In the present study, we do not have access to “other comorbidity” data but the inclusion of comorbidities would be an important consideration forthe design of future studies.
- Is there any relationship between mutation in CLL patients and adhesion phenotypes? This means certain mutations make more adhesive cells, as we saw in patients 19 and 166. If it does show, please elaborate in the discussion section. Thank you for your insightful comment. Analysing the adhesion data in Figure 1B, certain patients (e.g., #93 and #54) exhibit higher adhesion, with more CLL cells attaching to stromal cells compared to patients #88, #172, #182, and #183. Statistical analysis and correlation trends suggest that IGHV mutations, particularly in combination with 13q;14 deletions, are associated with a greater adhesion capacity compared to other genetic profiles.
- Please add the limitations of this study in the last paragraph of the discussion.
Limitation was added to the discussion line 261 to 263
- Are there any co-culture fluorescent image authors taken for both normal and disease conditions, just like Figure S1b?
The primary aim of this study was to investigate adhesion using flow cytometry, and as such, we did not perform fluorescent microscopy imaging of the co-culture. However, to confirm cell adhesion, we inspected the cultures under a bright field microscope.

Reviewer 2 Report
Comments and Suggestions for Authors
This manuscript investigates the role of the 5-lipoxygenase (5-LOX) pathway inhibitors in modulating stromal adhesion in chronic lymphocytic leukemia (CLL). It uses an ex vivo co-culture model to explore how genetic and clinical factors influence CLL-stromal cell interactions and evaluates the therapeutic potential of zileuton and MK886. The findings are significant for understanding alternative treatment strategies for CLL, but the manuscript has issues with clarity, grammatical accuracy, and referencing that need to be addressed.
Comments by Section and Line
Abstract
Line 18: Replace "leads us to investigate" with "motivated us to investigate" for conciseness.
Line 28: Provide references to support the conclusion that the 5-LOX pathway is a promising therapeutic target in CLL.
Introduction
Line 37: Provide updated statistics on cancer in general as well as this cancer type prevalence, including survival rates, to highlight the critical need for prognostic biomarkers. Cite “Cancer statistics, 2024, 2024”. Then give intro in cancer therapy in general, cite NIH paper “Cancer treatments: Past, present, and future, 2024” (PMID: 38909530)for more information. Add a citation to substantiate the claim that CLL shows heterogeneous clinical progression.
Line 45: Replace "various types of stromal cells to which they can adhere" with "various stromal cells that provide survival signals."
Line 48: Provide references discussing unmet clinical needs in double-refractory CLL cases.
Line 57: Add citations for the involvement of 5-LOX in inflammation and allergy responses.
Line 61: Expand the statement on 5-LOX's role in migration and adhesion with supporting studies. Mention previous studies using Chronic Myeloid Leukemia cell line, such as “Progesterone decreases viability and up regulates membrane progesterone receptors expression on the human Chronic Myeloid Leukemia cell line, 2024”
Results
Subsection 2.1:
Line 71: Clarify the rationale for using a lymphocyte count threshold of 30 × 10^9/L.
Line 89: Include a reference for the PCA methodology applied to support this analysis, such as previous study “Exploring the mechanism underlying hyperuricemia using comprehensive research on multi-omics, 2023”
Subsection 2.2:
Line 94: Replace "flow cytometry as detailed for one of the samples" with "flow cytometry as exemplified in Supplementary Figure S1." Cite previous Studies also using flow cytometry to support the approach, such as “The Role of Transient Receptor Potential Melastatin 7 (TRPM7) in Cell Viability: A Potential Target to Suppress Breast Cancer Cell Cycle, 2020,Effects of local anesthetics on breast cancer cell viability and migration, 2018”
Line 99: Include specific reasons for using the JeKo-1 MCL cell line as a positive control.
Subsection 2.3:
Line 127: Provide a citation for the mechanism by which ibrutinib disrupts CLL-stromal cell adhesion.
Line 139: Clarify the term "moderately strong correlation" with r-values and their significance thresholds.
Subsection 2.4:
Line 160: Replace "Notably, a cluster" with "A notable cluster."
Line 179: Specify the statistical significance of correlations between adhesion and mutation variables.
Discussion
Line 209: Add a reference for the known heterogeneity of CLL mutation status.
Line 225: The phrase "might lead to a higher cancer-cell burden" could be rephrased for clarity.
Line 231: Include a citation for the impact of TP53 deletion on CLL progression.
Line 253: Replace "merit further study to delineate" with "warrant further study to clarify."
Line 258: Provide additional context on the clinical approval of zileuton and its potential for repurposing.
Materials and Methods
Line 271: Add a reference to support the use of RosetteSep for CLL cell isolation.
Line 286: The washing steps in the cell-cell binding assay need to be explained more clearly.
Line 297: Specify the software versions for statistical analysis (e.g., ggplot2, FlowJo).
Author Response
The authors thank the reviewer for their time and valuable comments, which will help improve our manuscript. We have addressed the reviewer’s questions and comments as outlined below
Abstract
- Line 18: Replace "leads us to investigate" with "motivated us to investigate" for conciseness.
Thank you for your suggestion. We have made the change line 21.
- Line 28: Provide references to support the conclusion that the 5-LOX pathway is a promising therapeutic target in CLL.
Thank you for your comment. The role of the 5-LOX pathway in mantle cell lymphoma (MCL) and diffuse large B-cell lymphoma (DLBCL) migration and adhesion to lymph node niches has been well established. As According to the journal's guidelines, we are unable to include references in the abstract, instead we have included relevant references (references 30 to 32) in the introduction of the manuscript to support this.
Introduction
- Line 37: Provide updated statistics on cancer in general as well as this cancer type prevalence, including survival rates, to highlight the critical need for prognostic biomarkers. Cite “Cancer statistics, 2024, 2024”. Then give intro in cancer therapy in general, cite NIH paper “Cancer treatments: Past, present, and future, 2024” (PMID: 38909530) for more information. Add a citation to substantiate the claim that CLL shows heterogeneous clinical progression.
Text was added to the introduction explaining survival and incidence rates, lines 38-39 reference 2. For cancer treatment perspective, text was added lines 43 to 45 including references 4, 7-8 as per the reviewer’s comments.
- Line 45: Replace "various types of stromal cells to which they can adhere" with "various stromal cells that provide survival signals."
Thank you for your suggestion. We have made the change line 51.
- Line 48: Provide references discussing unmet clinical needs in double-refractory CLL cases.
Reference was added as per the reviewer’s comment, number 23
- Line 57: Add citations for the involvement of 5-LOX in inflammation and allergy responses.
Reference was added as per the reviewer’s comment, number 25
- Line 61: Expand the statement on 5-LOX's role in migration and adhesion with supporting studies. Mention previous studies using Chronic Myeloid Leukemia cell line, such as “Progesterone decreases viability and up regulates membrane progesterone receptors expression on the human Chronic Myeloid Leukemia cell line, 2024”
More information was added and expanded on lines 64 to 67 describing the findings reported in references 30 to 32
Results
Subsection 2.1:
- Line 71: Clarify the rationale for using a lymphocyte count threshold of 30 × 10^9/L.
This is explained in the results section, line 81. In order to obtain sufficient cells for the experiments, threshold level of lymphocyte count was arbitrary chosen.
- Line 89: Include a reference for the PCA methodology applied to support this analysis, such as previous study “Exploring the mechanism underlying hyperuricemia using comprehensive research on multi-omics, 2023”
Thank you for the suggestion. The explanatory text line 90 and reference 34 were added to support the methodology
Subsection 2.2:
Line 94: Replace "flow cytometry as detailed for one of the samples" with "flow cytometry as exemplified in Supplementary Figure S1." Cite previous Studies also using flow cytometry to support the approach, such as “The Role of Transient Receptor Potential Melastatin 7 (TRPM7) in Cell Viability: A Potential Target to Suppress Breast Cancer Cell Cycle, 2020,Effects of local anesthetics on breast cancer cell viability and migration, 2018”
Thank you for the suggestion. We have made the change line 102 as per the reviewer’s comment.
Moreover, the explanatory text lines 102 to 103 and reference 35 were added to support the methodology
- Line 99: Include specific reasons for using the JeKo-1 MCL cell line as a positive control.
Citations were added to explain the reason (references 36 and 37) on line 104.
Subsection 2.3:
- Line 127: Provide a citation for the mechanism by which ibrutinib disrupts CLL-stromal cell adhesion.
Thank you for the comment, References 38-40 was added.
- Line 139: Clarify the term "moderately strong correlation" with r-values and their significance thresholds.
The p-values wad added line 144. Moreover, the corelation coefficient values (r) were added for clarity on lines 150, 192,193,194 and 199.
Subsection 2.4:
- Line 160: Replace "Notably, a cluster" with "A notable cluster."
We have made the change line 177 as recommended
- Line 179: Specify the statistical significance of correlations between adhesion and mutation variables.
To clarify, supplementary table 1 has been added to the supplementary section, and p-values have also been included in the text. Lines 192,193,194 and 199
Discussion
- Line 209: Add a reference for the known heterogeneity of CLL mutation status.
Reference 43 was added
- Line 225: The phrase "might lead to a higher cancer-cell burden" could be rephrased for clarity.
The phrase is speculative, and we have clarified in line 233
- Line 231: Include a citation for the impact of TP53 deletion on CLL progression.
References 14, 46 and 47 was added
- Line 253: Replace "merit further study to delineate" with "warrant further study to clarify."
We have made the change lines 256-257 as recommended
- Line 258: Provide additional context on the clinical approval of zileuton and its potential for repurposing
References 49 and 50 was added
Materials and Methods
- Line 271: Add a reference to support the use of RosetteSep for CLL cell isolation.
Reference 51 was added
- Line 286: The washing steps in the cell-cell binding assay need to be explained more clearly.
The steps were clarified lines 288 to 289
Line 297: Specify the software versions for statistical analysis (e.g., ggplot2, FlowJo).
Versions were added (flow line 295) and ggplot2 (line 303)

Round 2
Reviewer 2 Report
Comments and Suggestions for Authors
good
Author Response
Dear Academic Editor,
In this study, we utilized a co-culture system as our sole method, which we have mentioned in the abstract. Additionally, we performed some analyses. We have elaborated on these procedures in the Methods section.